# Patients' experiences and perceptions of Guillain-Barré syndrome: A systematic review and meta-synthesis of qualitative research

Despina Laparidou[1], Ffion Curtis[2], Joseph Akanuwe[1], Jennifer Jackson[3‡], Timothy L. Hodgson[4‡], A. Niroshan Siriwardena[1‡]*

1 Community and Health Research Unit, School of Health and Social Care, University of Lincoln, Lincoln, Lincolnshire, United Kingdom, 2 Lincoln International Institute for Rural Health, University of Lincoln, Lincoln, Lincolnshire, United Kingdom, 3 Lincoln International Business School, University of Lincoln, Lincoln, Lincolnshire, United Kingdom, 4 School of Psychology, University of Lincoln, Lincoln, Lincolnshire, United Kingdom

☉ These authors contributed equally to this work.
‡ JJ, TLH and ANS also contributed equally to this work.
* nsiriwardena@lincoln.ac.uk

**Data Availability Statement:** All relevant data are within the manuscript and its Supporting Information files.

## Abstract

### Background

Guillain-Barré syndrome (GBS) is an immune-mediated polyradiculoneuropathy, with an incidence of 1-2/100,000 per year. Its severity is variable, ranging from very mild cases with brief weakness to severe paralysis, leading to inability to breathe independently, or even death. Currently there is limited evidence exploring the experiences of GBS patients. The aim of this study was to review patients' experiences and perceptions of GBS and its variants at diagnosis, discharge and during recovery, by conducting a systematic review and thematic meta-synthesis of qualitative studies of patients' experiences of GBS (and its variants).

### Methods

We searched twelve electronic databases, supplemented with internet searches and forward and backward citation tracking from the included studies and review articles. Data were synthesised thematically following the Thomas and Harden approach. The CASP Qualitative Checklist was used to assess the quality of the included studies of this review.

### Results

Our search strategy identified a total of 5,282 citations and after removing duplicates and excluding citations based on title and abstract, and full-text screening, five studies were included in the review and meta-synthesis; all included studies were considered of acceptable quality. Through constant discussions and an iterative approach, we developed six analytical themes following a patient's journey from suspecting that they had a health problem, through to being hospitalised, experiencing ongoing difficulties, slowly recovering from GBS, adjusting to their new circumstances, and re-evaluating their lives.

**Funding:** This systematic review is part of a study funded by a grant from the GAIN (Guillain-Barré & Associated Inflammatory Neuropathies) charity, United Kingdom (https://gaincharity.org.uk/). The funders had no role in study design, data collection and analysis, decision to publish, or preparation of the manuscript.

**Competing interests:** The authors have declared that no competing interests exist.

## Conclusions

Despite the variety of experiences, it was evident from all included studies that being diagnosed with and surviving GBS was a life-changing experience for all participants.

## Trial registration

Protocol was registered (CRD42019122199) on the International Prospective Register of Systematic Reviews (http://www.crd.york.ac.uk/PROSPERO).

## Background

Guillain-Barré syndrome (GBS) is an immune-mediated polyradiculoneuropathy, with an incidence of 1-2/100,000 per year [1]. GBS can affect anyone and at any age, although it is more frequent in adults and older people [2]. It has a highly variable onset, clinical severity and course, and there are several variants of GBS (including chronic inflammatory demyelinating polyradiculoneuropathy and Miller-Fisher syndrome) [3]. In most cases, GBS has an acute (4 hours) to subacute (up to 1 week) onset with symmetrical weakness and numbness of the limbs progressing proximally, usually over 2–4 weeks, causing loss of reflexes [3]. Although most patients recover (70% eventually experience full recovery), for some recovery can be slow or incomplete; for example, about 30 percent of individuals diagnosed with GBS have residual weakness after 3 years and about 15 percent experience long-term weakness [2]. So even though GBS is not considered a chronic condition, it often has long-term effects and patients may have ongoing neurological deficits that affect their quality of life, their work and social lives [4, 5].

Currently there is limited evidence exploring the experiences of individuals who have had GBS. Previous research highlights the significant impact of GBS on individuals and their families. One study [6] exploring the presence of residual symptoms and the long-term effect of GBS on patients' daily lives, working activities, hobbies and social status, showed that, at follow-up, most patients had made a complete functional recovery; however, almost a third had to make substantial changes in their daily lives including jobs, hobbies or social activities. Similarly, Khan and colleagues [7], examining factors impacting long-term health-related outcomes in GBS survivors, found that it had moderate to extreme impact on ability to participate in work, family, and social activities for 16% of participants. GBS also had a substantial impact on mood, confidence and ability to live independently for 22% of their sample, while moderate to extreme depression, anxiety and stress were also reported. Another study [8] exploring utilization of and satisfaction with healthcare resources, informal help and the burden of care on family caregivers during the first 2 years after onset, found that although most participants were satisfied with their overall care, they were less satisfied with the information they received about GBS or with the cost of care. At 2 years after onset, almost a third of participants still depended on informal carers and their spouses, often expressing increased concern and responsibility for their household and family. Finally, a systematic review [9] of the literature on GBS patients' quality of life after onset of the disease concluded that many patients felt limited by their condition, even years after the onset, and that GBS had a lasting psychosocial impact, affecting patients' mental well-being, daily activities and work life.

### Theoretical perspective

We used two theoretical models to facilitate data gathering, analysis and interpretation of the experiences of patients with GBS, the Illness Trajectory Framework (ITF) [10] and Taylor's [11] theory of cognitive adaptation to threatening events.

According to the ITF [10], chronic illnesses follow a course or *trajectory* which, although different for each individual, have eight common *trajectory phases*, which involve changes in health and the type of interventions or *trajectory scheme* needed: pre-trajectory, trajectory onset, crisis, acute, stable, unstable, downward, and dying. Overall, the ITF argues that chronic illness (and management) has an effect on different aspects of individual's identity, forcing patients to make identity adaptations to live (*come to terms*) with their chronic condition, its consequences and their own mortality.

The [11] theory of adaptation can be viewed as complementary to ITF, focusing particularly on the individual's cognitive response when faced with a threat. According to Taylor [11], when individuals are faced with a threatening event, such as a serious illness like GBS, they go through a readjustment process, during which they try to understand why the event happened, what caused it, and how it has affected their lives. Patients also try to regain mastery over their illness, while avoiding something similar happening again, as well as find ways to feel good about themselves again and enhance, or restore, their self-esteem. Often, making favourable social comparisons is the key to such enhancing attempts to regain control, whereby patients try to cope by feeling better off than others in the same situation, thus, making more positive self-evaluations and ultimately feeling better adjusted and able to cope with their illness.

The findings of this review will provide insights into the patient journey, which could be useful in informing patient care and support services.

## Aim of the study

Given the severity of GBS and how variable its onset and course are, it is imperative to explore in depth the patients' experiences of GBS at the time of diagnosis, during their hospitalisation, in the period post-discharge from hospital and on their return to the community, in order to better understand this patient group's health and social care needs, as well as explore any potential facilitators and barriers to recovery and return to function. Furthermore, based on our preliminary searches, to date no systematic reviews of qualitative studies exploring patients' experiences of GBS have been published.

The aim of this study was to review patients' experiences and perceptions of GBS and its variants at diagnosis, discharge and during recovery, by conducting a systematic review and thematic meta-synthesis of qualitative studies of patients' experiences of GBS (and its variants).

## Methods

We followed ENTREQ guidelines for enhancing transparency in reporting the synthesis of qualitative research [12]. The review protocol was registered with the PROSPERO International prospective register of systematic reviews [13] and is available from: http://www.crd. york.ac.uk/PROSPERO/display_record.php?ID=CRD42019122199.

Our review question was: What are patients' experiences and perceptions of GBS and chronic inflammatory demyelinating polyneuropathy (CIDP) and its care at diagnosis, discharge and during recovery?

## Inclusion criteria

Studies were eligible for inclusion if they had a qualitative research design (e.g. interviews, focus groups, ethnography) and reported on patients' lived experience of GBS and CIDP. Healthcare services (including the treatment, care and support they provide) have changed considerably in the last 20 years and consequently so has patient experience of care. To ensure that the reviewed patient experiences are relevant to the present day and can inform

improvements in current practice, only studies published between January 2000 and May 2020 were eligible for inclusion. In addition, only peer reviewed studies, written in English were considered for eligibility.

Qualitative studies published outside these dates or in other languages were excluded. Quantitative studies were also not eligible for inclusion, since we were interested in patients' lived experience of GBS and CIDP and we wanted to include in depth accounts of their experiences (preferably expressed in their own words, i.e. by using quotes).

## Information sources and search strategy

Electronic database searches were performed in the Cochrane Library, Joanna Briggs Institute Evidence-Based Practice Database, PROSPERO, MEDLINE, Academic Search Complete, AMED, CINAHL, Humanities International Index, PsycARTICLES, PsycINFO, EMBASE, and PubMed. All databases searches were supplemented with internet searches (i.e. Google Scholar), and forward and backward citation tracking from the included studies and review articles.

The search strategy used in all the above databases included a combination of two sets of keywords and related terms: 1) Guillain-Barré syndrome (GBS), chronic inflammatory demyelinating polyneuropathy (CIDP), acute inflammatory demyelinating polyneuropathy (AIDP); combined with 2) qualitative research, interview, focus group, experiences, perceptions, attitudes, and views. The search terms were entered using Boolean operators and truncation, wherever deemed necessary. Medical Subject Headings (MeSH) were also employed in forming the search strategy. For the full search strategy used for the Medline database, see Table 1.

## Study selection and data extraction

All references were reviewed and screened by three reviewers (FC, DL, JA) independently. Titles and abstracts were initially screened for relevance and final eligibility was assessed through full-text screening against the inclusion criteria, using a pre-designed study selection form. Any disagreement between the reviewers over the eligibility of references was resolved through discussion between the reviewers, and in consultation with a fourth reviewer (ANS) when necessary.

**Table 1. Search strategy for MEDLINE database.**

| Search ID | Search terms |
|---|---|
| S1. | (MH "Guillain-Barre Syndrome") OR (MH "Miller Fisher Syndrome") OR (MH "Posterior Cervical Sympathetic Syndrome") OR "Guillain–barré syndrome" |
| S2. | Guillain-Barre syndrome or gbs or Guillain-Barré |
| S3. | (MH "Polyradiculoneuropathy, Chronic Inflammatory Demyelinating") OR (MH "Guillain-Barre Syndrome") OR (MH "Polyneuropathies") OR (MH "Demyelinating Diseases") OR "chronic inflammatory demyelinating polyneuropathy (cidp)" |
| S4. | S1 OR S2 OR S3 |
| S5. | (MH "Qualitative Research") |
| S6. | interview* or focus group* |
| S7. | experience* or perception* or attitude* or view* or feeling* or opinion* or reflection* or belief* |
| S8. | acute inflammatory demyelinating polyneuropathy OR aidp |
| S9. | S1 OR S2 OR S3 OR S8 |
| S10. | S5 OR S6 OR S7 |
| S11. | S9 AND S10 |

A standardised, pre-piloted form was used to extract data from the included studies for assessment of quality and data synthesis. Extracted information included: study details (title, authors, date, country), methods (aims, objectives, research questions, study design, setting, data collection methods), participant characteristics (demographics, inclusion/exclusion criteria, method of recruitment, sample selection, sample size), and study findings (main and secondary outcomes, data analysis, conclusions). One reviewer extracted data and a second reviewer checked the data extractions for accuracy. Any discrepancies were resolved through discussion and missing data were requested from study authors.

## Data synthesis

Data were entered into NVivo 11 qualitative data analysis software to facilitate analysis. We used thematic synthesis to synthesise the data, following the Thomas and Harden [14] approach. Initially, three reviewers (DL, FC, JA) independently coded the 'results' sections (and 'discussion' sections, where new concepts were introduced) of the included papers line-by-line, according to meaning and content, using an inductive approach. Consequently, these free codes of findings were organised into 'descriptive' themes that encompassed the meaning of groups of the initial codes. Finally, based on the codes and 'descriptive' themes and through discussion with the wider review team, the final 'analytical' themes were developed.

We followed an inductive approach to analyse and synthesise the data, rather than imposing the illness trajectory framework (ITF) or Taylor's theory of cognitive adaptation onto our results. Instead, we used both these models to interpret the results and describe the patient journey from experiencing the first GBS symptoms to hospital discharge and recovery (see Discussion below).

## Quality assessment of studies

The Critical Appraisal Skills Programme (CASP) Qualitative Checklist [15] was used to assess the quality of the included studies of this review. Low quality, however, was not a criterion for exclusion of a study, since we were interested in the synthesis and interpretation of all relevant and sufficiently rich data. The CASP qualitative checklist aims to assess various elements of qualitative research studies, including research aims, appropriate methodology, research design and strategy, methods of data collection and communication between researchers and participants, ethical considerations, rigor of data analysis, and the clarity and value of study findings.

Three reviewers (JA, FC, DL) independently assessed the quality of the included studies. Discrepancies were resolved by discussion and consensus, and in consultation with a fourth reviewer when needed (ANS).

## Reflexive statement

Reflexivity enables authors to recognise the assumptions and preconceptions they bring into the research and may influence the research process, while allowing the reader to understand the dynamics between the researcher and the researched. This review was commissioned and developed in discussion with the chief executive and chair of the board of trustees of the charity 'Guillain-Barré and associated Inflammatory Neuropathies' (GAIN). GAIN was the main funder of the study and helped develop the review's main objectives: exploring GBS patients' experience of care, particularly focussing on the period following discharge from hospital and return to the community.

The reviewers (except for ANS and JA) had no prior knowledge of or experience with GBS. DL, a psychologist by background and a researcher in health services, has experience in

quantitative systematic reviews and in the analysis of qualitative data. FC is a research fellow whose research predominantly focuses on non-pharmacological interventions for the prevention and management of chronic conditions. She has experience conducting systematic reviews of both quantitative and qualitative studies. JA has a background in clinical nursing and public health with expertise in qualitative and quantitative research methods, and systematic reviews. As a nurse, JA has general clinical knowledge of GBS. JJ is a community researcher with interest in the patient and service user experience, as well as extensive experience of qualitative and engaged research within the public sector, community and voluntary groups. TLH is a psychologist with interest in cognitive deficits in patients with neurodegenerative disorders, with experience of analysis of quantitative data and knowledge of the neurology of cognition and perception. ANS is a clinical academic general practitioner (GP) by background with expertise in social science methods, including systematic reviews, qualitative meta-syntheses and qualitative studies more generally. He has general clinical expertise and insight into GBS but is not an expert on the condition itself.

Familiarisation with the papers included in this review, together with being aware of the existence of GBS-dedicated charities, may have influenced the authors' suggestions regarding potential sources of support for patients in the future; such support may be available from multiple sources, and our view may have been influenced by the funding for this study.

## Results

The search strategy identified a total of 5,282 citations. After removing duplicates and excluding citations based on title and abstract, 63 articles remained for full-text screening. A further 58 articles were excluded based on inclusion/exclusion criteria (main reasons for exclusion: paper not written in English, and/or a quantitative design), leaving five studies to be included in the review and meta-synthesis. Fig 1 presents a flowchart illustrating the results of the selection process.

### Characteristics of included studies

The five included studies (Table 2) were published between 2003 and 2015 and were from four countries: Australia [16], Sweden [17, 18], UK [19], and, USA [20]. One study [20] was a PhD thesis. All studies interviewed people living with GBS; none of the studies included participants diagnosed with other variants of GBS, such as CIDP or Miller-Fisher syndrome. All studies contained both men and women, but there were more male participants overall (55/94). Participants' ages ranged from 16 to 80 years. Studies were based on individual semi-structured interviews only; none included focus groups. Various methods of analysis were employed in the studies, including content analysis [17, 18, 20], interpretative phenomenological analysis [19], and the constant comparative method [16]. Most of the studies focused on the overall experiences of people during hospital care or recovering from GBS, whereas one study [19] focused specifically on their experiences returning to work following recovery from GBS.

### Quality assessment of studies

Table 3 presents the results of the critical appraisal of the five studies, using the CASP criteria for qualitative research. All included studies were considered of acceptable quality. It should be noted here that three [16, 17, 20] of the five studies performed less well on the reflexivity question (CASP 06) and one study [17] performed less well on four CASP questions, since not enough relevant information was reported in the papers.

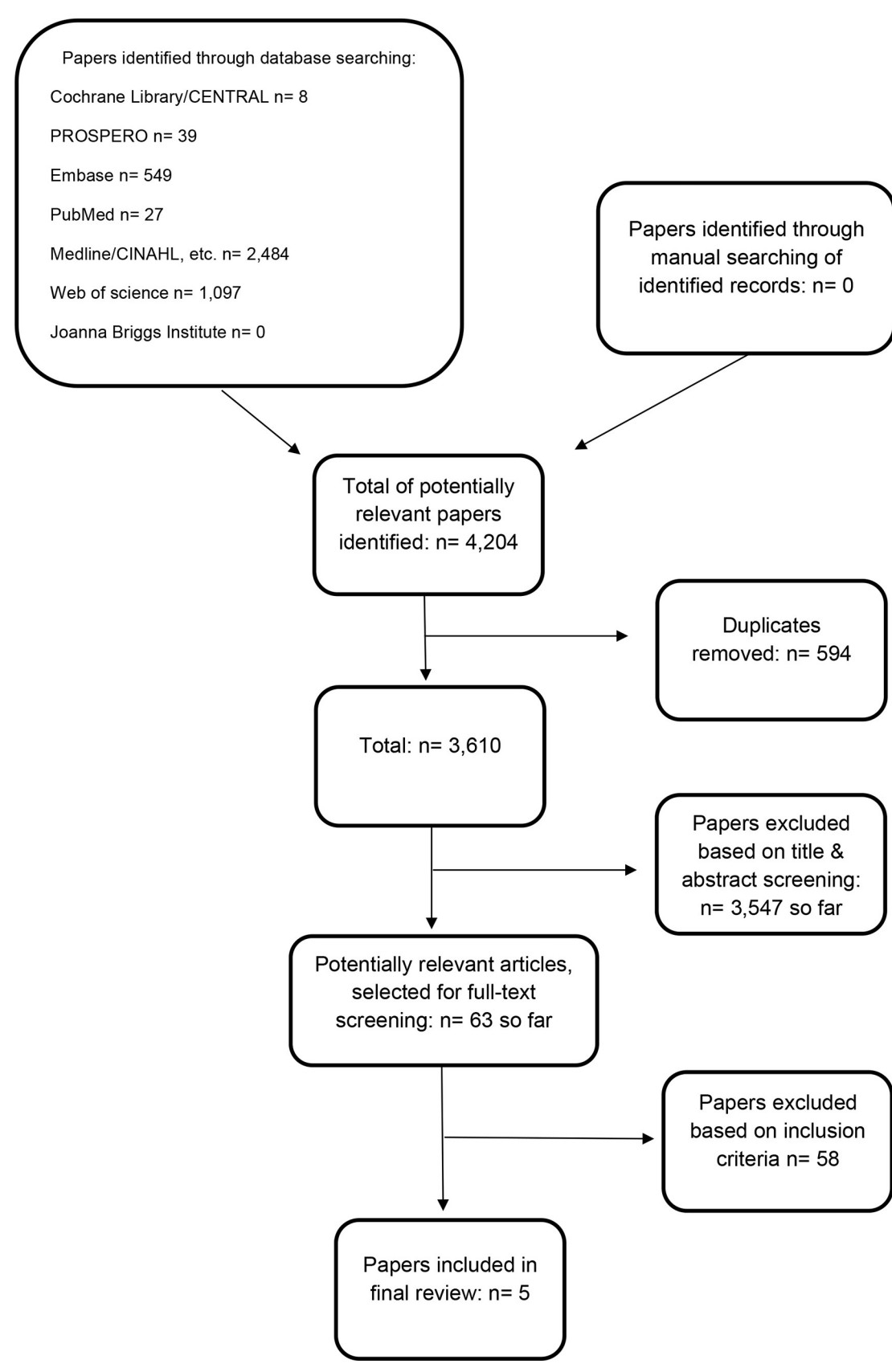

**Fig 1. Flow diagram of study selection.**

**Table 2. Study characteristics.**

| Study | Study aims | Sample | Method of data collection | Method of data analysis |
|---|---|---|---|---|
| Cooke & Orb, 2003; Australia | To "examine the perspectives of patients with Guillain-Barré syndrome during their recovery phase" | Purposive sampling of 5 participants (3 male, 2 female) admitted to the hospital with a diagnosis of GBS; discharged from hospital in the last 2 years<br><br>Ages: 28–67 years | Semi-structured individual interviews with open-ended questions | Constant comparative method |
| Forsberg et al., 2008; Sweden | To "describe experiences of falling ill with GBS, with the focus on the onset of disease, the diagnosis and the illness progress during hospital care" | The study population (35 participants: 22 male, 13 female) was identified in a previous multicentre study, including eight hospitals. Participants were approached 2 years after illness onset.<br><br>Ages: 20–78 years | Individual interviews | Content analysis |
| Forsberg et al., 2015; Sweden | To "describe experiences of disability in everyday life and managing the recovery process two years after falling ill with Guillain-Barré syndrome" | The study population (35 participants: 22 male, 13 female) was identified from a previous longitudinal study. Participants were approached 2 years after illness onset.<br><br>Ages: 22–80 years | Semi-structured individual interviews | Content analysis |
| *Hooks, 2015; USA | To "gain a richer understanding of the patient's recalled experience of an acute episode of moderate to severe Guillain-Barre' syndrome" | Individuals, from eight different states, with a prior self-identified diagnosis of moderate to severe GBS. The sample (recruited through different strategies) included 10 females and 4 males.<br><br>Ages: 19–79 years | Semi-structured individual interviews | Content analysis |
| Royal et al., 2009; UK | To "investigate in greater detail participants' motivations for returning to work after GBS, the subjective impact of GBS on their ability to return to work, and how they managed their return to work." | Five GBS patients (4 male, 1 female) who had returned or attempted to return to work following their illness. Participants were selected from a database of patients, who had been admitted to a specialist rehabilitation unit between 2000 and 2006. Participants were interviewed between 1 and 5 years after their admission to hospital with GBS.<br><br>Ages of onset of GBS: 25–63 years | Semi-structured individual interviews | Interpretative phenomenological analysis |

* PhD thesis.

## Data synthesis

After initial coding and development of descriptive themes, we developed six analytical themes (Table 4). We organised the themes chronologically into a model representing a patient's journey from their initial suspicion that they had a health problem, through to being hospitalised, experiencing ongoing difficulties, slowly recovering from GBS, and resuming their everyday lives.

**From uncertainty to hope.** Participants tried to ignore initial strange sensations due to GBS [17, 20]. As the condition deteriorated, they attempted to explain their symptoms as the

**Table 3. Quality assessment of studies.**

| Study | CASP01 | CASP02 | CASP03 | CASP04 | CASP05 | CASP06 | CASP07 | CASP08 | CASP09 | CASP010 |
|---|---|---|---|---|---|---|---|---|---|---|
| Cooke & Orb, 2003 | | | | | | | | | | |
| Forsberg et al., 2008 | | | | | | | | | | |
| Forsberg et al., 2015 | | | | | | | | | | |
| Hooks, 2015 | | | | | | | | | | |
| Royal et al., 2009 | | | | | | | | | | |

**Table 4. Analytical and descriptive themes.**

| Analytical theme | Descriptive themes | Some supporting quotations |
|---|---|---|
| Theme 1: From uncertainty to hope | Initial strange sensations | "Most participants described the initial symptoms that were manifested as strange or odd sensations or peculiar feelings" [20]. |
| | Rationalising symptoms & misattributing diagnosis | "A few persons tried to ignore the strangeness of their bodies, but others came up with explanations such as being tired or overworked. [. . ..] Some also described a fear of having a better-known disease such as cancer or multiple sclerosis" [17]. |
| | | "This participant went to the physician's office for care. The physician's office called for emergency support and transport because they felt this participant was having a stroke" [20]. |
| | Participants' eagerness to find out what's happening to them | ". . .when I got to the hospital, [I felt] more curious. What's going on? What's causing this?" [20] |
| | Uncertainty | "The uncertainty was overwhelming for many and affected their whole lives" [17] |
| | Healthcare professionals' lack of knowledge and experience with GBS | "I just wish that, um, that the people who were treating me had known more about it, You know, it's not comforting when, you know, a nurse walks in and said: I didn't know anything about it, so I had to Google it. You know, I didn't find that comforting. And there were–there was more than one person that– that said they had never really heard of it" [20] |
| | Need for information about GBS | "Many participants expressed the desire to have more information about their illness. In addition, participants commented on the value of this knowledge and the impact that it had in terms of their future outlook" [20]. |
| | Friends and family the main source of information about GBS | "My daughters–my three daughters flew in, and they were there with my wife. And, uh, they are all very techy, so they were looking up everything they could find on the Internet about it at the time" [20]; |
| | | ". . .another participant discussed the value of the resources provided by the GBS/CIDP Foundation International" [20] |
| | Prospect of a positive prognosis | "The concern of having a very serious disease and the fact of a prolonged recovery was becoming a realization, while others in the same situation still continued to rely heavily on the prospect of a positive prognosis" [17] |
| | Hope of recovery | "After being confronted with physical dependency and then encountering helplessness, the participants were hopeful that recovery was near" [16]. |
| Theme 2: Feeling lost in a changing life | Experience of physical symptoms | "In the morning I felt numbness in one hand, and in the afternoon I could not swallow. By midnight, I could not breathe and was taken to intensive care for ventilator support" [17]. |
| | Loss of identity | "Some talked about being handicapped and losing their identity as an independent person" [17] |
| | Dependency, vulnerability and feelings of helplessness | "Their bodily limitations disappointed them and made them feel helpless, as they were dependent on assistive devices or were not able to participate in activities in the same way as before" [18] |
| | Feelings of shame and embarrassment | ". . .when I could still walk they bathed me in the shower, which I thought was horribly traumatizing. Because, you know, I was 25. And being bathed by someone was, like, extremely embarrassing. So that was, I think, the first day of, like, my traumatic–when I say my traumatic experience. When I started crying like every day. That was the first day. . .maybe a day or two later is when I couldn't walk. So then they came in and gave me a sponge bath. So I ranked that as even worse. It's just very–It felt very demeaning because, you know, I was still young and somebody here bathing me on my bed. I thought it was terrible" [20] |
| | Psychological responses to GBS | ". . .there were so many. Frustration that I couldn't figure out what was wrong with me. Guilt because I was taking so many drugs. There was. . .a not knowing what was wrong with me was, um, just heart-breaking. A frustration when I fell at work, I laid in the doorway of my job, and I just cried"." This variety of emotions is mirrored in another participant's quote as well: ". . .Elation, I guess. When you're finally going to get out of the hospital. . .Anger over the care at rehab. And anger with the doctors not being able to listen. . .doctors are amazingly poor listeners. And gratitude. I mean, geez. I don't know. You know, it's one of those times where thanks is not adequate, but it's all you got" [20] |

*(Continued)*

**Table 4.** (Continued)

| Analytical theme | Descriptive themes | Some supporting quotations |
|---|---|---|
| | Effects of GBS on family life | "I was worried about my husband too, I guess. That was a part of it. All my family–I'm from a big family, but everybody lives someplace else. . .But as I got better, I worried about him and, you know, where he was getting support" [20] |
| | | "This was the first Christmas in 38 years she ever prepared a Christmas ham. All these years it was I who'd done it–done all that kind of thing" [18] |
| | Difficulties with re-assuming social lives | "The social relationships and friendships you've had before get curtailed somehow. I just don't get together with a lot of my friends–friends I met through memberships in clubs and recreational groups" [18] |
| | Experiencing work-related difficulties | "So some days when I phoned in and said I'm not coming in today it wasn't necessarily because my legs weren't working properly or I didn't feel strong enough physically, it was mentally I didn't feel strong" [19] |
| Theme 3: Fractured care | Lack of continuity of care | "Lack of continuity was also described, for example having to wait a long time for referrals that evidently delayed the start of rehabilitation" [18]; "The fact that my legs were so sensitive, and it felt like my legs were always either hot or cold; and that's not their fault, but there weren't enough nurses to keep ice packs and heating packs coming. . .So just limited staff. . .And that really saved me was when I get an ice pack, I felt like everything was going to be okay" [20] |
| | Lack of person-centred care at hospital | "There were some [staff] during the day, and I knew who to ask for to help me do that during the day. But in the evening, they basically told me they were not going to help me. That I weighed too much" [20]; |
| | | "I will take the bedside commode, but I will not do the bedpan. So that's really when they started telling me they were not going to move me" [20]. |
| | Feeling not listened to by healthcare staff | ". . .my worst experience about this was having the alarm on my bed. And you know hearing the voice that come over the intercom saying: Do not get out of bed! And to be yelled at. I mean that's the only way I can put it. You know they weren't really mean, but you know, being told, 'Don't get out of bed', you know, all of the time just kind of ticked me off a little" [20]; |
| | | "And anger with the doctors not being able to listen. . .doctors are amazingly poor listeners" [20] |
| | Communication issues with healthcare staff | "I mean, the doctors–honestly, I understand that, you know, you're doctors. You're busy in a big hospital. But they don't have time to sit down with you and explain exactly everything that is going on. It's just: This is what you have. This is sort of how it works: And here's your prognosis'" [20]. |
| | Feeling that needs are not being met by healthcare staff | "I remember one day they had me riding on a bicycle. . . [and] a physical therapists [should] not push, push, pushing somebody to the point of exhaustion. That's the only mistake–I wouldn't even say mistake. But you know what I mean. It's the only big thing. . .I just think it's set up for a different kind of rehab than what I needed" [20]; |
| | | ". . .the mental process of going through this illness was never addressed, really, except by me" [20] |
| | Lack of publicity about GBS | "A silly example, if somebody is pregnant, when they come back you know basically what to expect. . .and how to treat that person. . .but you can't blame the management, you can't blame work colleagues because [people] don't know. . ." [19] |
| Theme 4: Positivity towards recovery | Positive feelings towards healthcare received (e.g. physical care needs were met; kind staff; medical staff were conscientious) | ". . .my hunch is that they probably didn't have a lot of experience with it. . .. they might not have been prepared. . .but as far as dealing with my disabilities, they were great. Even if they didn't understand what caused it themselves. But, you know, to help me do the necessities" [20]; |
| | | "I still remember, when I am getting myself dressed, there was an aid. . .and she was the one that would help me get dressed. So she, you know would, I'd want to put my undies on and pull them up, and I'd want to put my pants on and pull them up. And she said. . .we're going to do this a better way. We're going to put those undies on and the pants on, then we're going to pull up one time. And I thought: Great idea!" [20] |

*(Continued)*

**Table 4.** (Continued)

| Analytical theme | Descriptive themes | Some supporting quotations |
|---|---|---|
| | Getting support from family and friends (including colleagues) | "I can turn myself but it's a challenge. But my wife helps me" [18]; |
| | | "That's the kind of thing I think is almost the hardest: when people say 'Well obviously you just have to try harder'" [18]; |
| | | "... I think it is important to get back on board, need the company and intellectual stimulation. Brain was starting to numb out a bit, definitely, and you know it's quite a close knit company that I work at. I consider them all to be friends as well, so there is a support structure there" [19] |
| | Importance of peer support | "There were two people who had GBS...One was this man. He had GBS and now he was working and he was back at work and we related and talked. And the other one was Miss L...She still calls me every now and then. Just came by to talk and visit with me. And that was very nice, and that was–and they told me a lot about...recovery, I guess. Recovery was possible, you know. That was very helpful...And it was sort of an affirmation that things should get better" [20]; |
| | | "If you ever have anybody in here, in this condition...I'd be happy to come over, and I'll shoot the breeze for a little while. And tell them, you know, you can get better, you know? And I think if I would have heard something like that from somebody who had been through it, rather than a caregiver giving it to me, I think it would have had a lot more impact on how I would have perceived everything. I really do. The issue [with hearing it from a caregiver] was probably a lack of credibility on my part" [20] |
| | Maintaining a positive attitude | "They told me that the course is positive for nine out of ten patients, so most recover completely. I got the feeling that now it is just to be strong and then everything will be fine" [17]; |
| | | "Your life isn't over. And I know it seems like it but it's not. It's going to get better but your attitude is everything. Your attitude will make or break this thing for you" [20] |
| Theme 5: Adjustment | Experiences of recovery varied | "There was a broad variation in the participants' experiences of the recovery process and adjusting to their new situation after falling ill with Guillain-Barré syndrome. Some said that they did not have any residual symptoms, and had regained full bodily strength and health by the time of the interview. They had experienced a recovery period lasting a couple of months, and gradually all their symptoms had resolved." [18] |
| | Symptom management | 'If you've asked for something, you get help. But as for the actual problem...you end up having to pretend you're some kind of expert on your own illness after a while, in order to really get the help you need.'" [18]; |
| | | "Strategies to lessen the impact of leg weakness included deliberately walking more slowly in order to save energy. 'If I speed up, I can't walk very far, but if I walk slowly, I can almost walk any distance'" [18]. |
| | Need for control and independence | "And then when I started to exercise and discovered that I could stand up, everything went very fast" [17] |
| | Achieving milestones | "A desire to move on and get this thing behind me. One of the best ways of doing that is going back to work, definitely" [19]; |
| | | "Well, little by little, you know things got better. I mean, I can remember being so excited at home when I had to–I could actually get myself a cup of coffee and walk across the room without a walker...with a cane...and sit down in a chair, and do that all by myself. I thought that was just . . .a major feat" [20] |
| | Acceptance of their situation | "The descriptions of the recovery process showed that the participants were in different phases of acceptance and coping, independently of their current health status. Some neglected the influence of the consequences of the disease on their life situation. Others said that they had reappraised their new situation and found ways to manage their restrictions and limitations in activities." [18]; |
| | | "It may not be the light that you're hoping to see, but at least, you know there's light" [20] |

(*Continued*)

**Table 4.** (Continued)

| Analytical theme | Descriptive themes | Some supporting quotations |
|---|---|---|
| | Remaining positive, despite persisting symptoms | "Much stiffer. . .you feel so old in the morning when you try to put your socks on. You have to put your foot on your knee. . .but it's not like I'm suffering from that" [18]; |
| | | "One participant who had to use a wheelchair said: 'I'm so relieved I didn't lose my speech or my mind '" [18]. |
| Theme 6: Towards a new self | Desire to go back to 'normal' pre-GBS selves | "I have one goal, and that is to get well. And I mustn't let myself think it's hard. As soon as I'm able to walk, we can get going on all the other stuff" [18]. |
| | Hiding GBS | ". . . I didn't want anybody to, if you like, know the truth about what I was, how I was feeling at the time" [19]. |
| | Fear of stigma | ". . . I must admit I didn't tell them I couldn't feel my feet." [19]; |
| | | "People look at you and think what's the matter with him? No honestly, work is okay now [but], when I first went back I did have to concentrate very, very hard because I didn't want to draw attention to myself" [19] |
| | Others' perceptions of GBS | "To start with I was really self-conscious and I still am a little, but not as much as I was. Colleagues actually seeing my lips don't move in the normal way and I know that sometimes I would say a word and I was quite sure that they knew what the word was but they got confused. . . they wonder what's going on, but no-one actually normally says anything" [19] |
| | Returning to work seen as going back to 'normal' self | "That [work] was good for me because I was normal again, I thought I was the same as I was before I had the Guillain- Barré, I was the same person again. Obviously I wasn't but I thought I was, and that was good enough for me" [19]; |
| | | "I want it all to be forgotten, as if it never happened. . . so I'd rather be back, I'm quite happy to get back into normal life" [19] |
| | Motivations for going back to work | ".... I ended up with about thirty pounds a week benefit and then I realised I had to try and get back to work as soon as possible" [19]; |
| | | "The job, it was probably therapeutic for me because I didn't use to go to the depot in my car. . . it gave me a mile walk, about three quarters of an hour walk down to the depot which I thought was therapeutic. . . it was therapy for me but I was getting paid for [it]" [19] |
| | Facilitators to going back to work | "I was able to drift off into the meeting room and have half an hour's kip on the sofa, which everyone was totally okay with. . . if you do that during the middle of your working day, it completely recharges your batteries" [19]; |
| | | "Friday is like a buffer day. . . I think that's quite important and basically you know, I can have a good sleep, have a breather and recharge my batteries and then the weekend is for the family and Monday is a buffer day similarly at the other end. . ." [19]; |
| | | ". . .so what I was able to do was actually go out and employ someone who was much more specific to the needs that I now have" [19] |
| | Barriers to going back to work | "You can't do that [rest] at work, when you get to work if you've had a bad journey, you know, you can't have a lie down unfortunately" [19]; |
| | | "Although most workplaces offered some flexibility at first, this was usually only possible on a short-term basis. [. . .] But only Andrew and Edward had sufficiently senior positions to resist taking all aspects of their job back, '. . . You've got to be careful not to take all that. . . back on. . .' (Andrew), '. . . I've said no' (Edward)" [19]. |
| | Re-evaluating life | "The participants also described positive aspects, in that the onset of disease and the recovery process had made them reflect on their life situation, and some had made significant changes in their work situation or choice of leisure activities" [18]. |

result of normal everyday activities or occurrences (such as medication side effects or tiredness), or feared having better known conditions, such as cancer [17, 19, 20]. On occasions, healthcare professionals either misdiagnosed their symptoms (e.g. thinking the participant was having a stroke) or felt they were feigning illness [20].

Uncertainty for participants often became overwhelming, making them eager to find out what was happening to them [17, 20].This feeling of uncertainty was made worse by a general lack of information and knowledge of GBS, among patients and healthcare professionals [16, 17, 20]. Most participants had never heard of GBS and felt that healthcare professionals were lacking knowledge of and experience with GBS, which they did not find comforting, and left them needing more information about their illness [20]. Often patients' families and friends that had to find information about GBS themselves, mainly through searching on the internet [16, 20]. Many participants would have liked to have received more information about GBS to improve their understanding of their condition, but also because they found information about GBS, especially on prognosis and recovery, to be reassuring [16, 17, 20].

Although some participants were sad and disappointed about the long recovery ahead of them, many relied on the prospect of a positive prognosis and hoped for a full recovery [17]; indeed, participants' hope of recovery was their main motivation, giving them the courage to continue [16–18].

**Feeling lost in a changing life.** Having GBS affected every aspect of participants' daily lives [16–20]. A variety of physical symptoms and problems were experienced at onset, starting with numbness or tingling in hands and feet, pain, leading to full paralysis requiring ventilation support [16–20]. Participants' experiences varied, but many had ongoing difficulties, such as fatigue and memory difficulties, that still limited them in their everyday activities even two years after the onset of their illness [16–20].

As a result, some participants felt they had lost their identity as an independent person [17, 19], while others felt helpless through loss of independence [16, 18]. These feelings were accompanied by shame or embarrassment, especially when help with hygiene was needed or appearance had altered as a result of GBS [17, 19, 20]. Participants described many emotions as a result of being diagnosed with and needing care for GBS, such as frustration, guilt, elation, anger, and gratitude [19, 20]. Other common psychological responses were: feeling lost; feeling abandoned; anxiety when in the intensive care unit or connected to a ventilator; sadness; depression; and fear [17–20].

Having GBS influenced participants' family lives. Participants were worried about their family's wellbeing, while they were hospitalised [20], or felt sad and frustrated about how limited mobility had forced them to accept changed family responsibilities [18]. Living with GBS made it harder for them to participate in society and restricted their social lives considerably, restricting visits to friends and travel [18]. Finally, another area affected by a participant's condition was their work, since physical restrictions and residual GBS symptoms affected function or prevented them from returning to work [18, 19].

**Fractured care.** Participants were generally dissatisfied with healthcare service support which they considered a key barrier preventing recovery from GBS. Participants felt that there was lack of personalised and person-centred care, lack of continuity of care, lengthy waits for referrals, and staff shortages, which made it harder for participants to receive the care they needed [18, 20].

Participants also felt not listened to by healthcare staff and experienced poor communication from healthcare professionals [18, 20]. They sometimes felt doctors' attitudes were 'cavalier' or healthcare professionals lacked time to discuss their condition with them properly [20]. Specific needs, whether physical and psychological, were often not met [20].

Participants also identified a want of publicity about GBS as a main factor contributing to the lack of support they received, especially when returning to work, as the public did not have any insight into the long term effects of GBS or what to expect when interacting with the participants [19].

**Positivity towards recovery.** In contrast, participants variously expressed an overall satisfaction with the care they received from community and hospital healthcare, commending kind staff attending to their physical care needs with efficient treatments [16, 18, 20]. More specifically, participants were often very satisfied with nursing care for their physical, psychological and social problems [16, 20].

Another facilitator to recovery from GBS was the invaluable support from their family and friends, including both practical (e.g. help with the home and transportation) and psychosocial support (e.g. gathering information about GBS or being emotionally supportive) [16, 18, 20]. However, in one study [18] participants expressed frustration over the lack of understanding from family and friends in relation to their physical limitations and the effect on the participants' capacity and everyday life in general.

Support from colleagues was also viewed very highly by participants and motivated them to go back to work, as they often considered their colleagues as friends as well [19]. Often, however, participants were ambivalent towards their co-workers, initially being grateful for their support, but finding them over-protective at the same time [19]. When this support soon decreased, participants found it difficult to perform their jobs, demonstrating how important such support had really been for participants [19].

Peer support was also viewed as really important by participants [16, 20], who valued being able to talk with and receive information from others who had also been ill with and survived GBS, as this communication filled them with hope about recovery and the future [20]. Equally, participants would have gladly done the same for others in their situation, as they considered peer support to be better received and more impactful [20].

Finally, participants' positive attitude was a major facilitator to their recovery from GBS and helped them realise that life wasn't over and that things would eventually improve [17, 20]. Hope and confidence in recovery were huge motivators, especially once their functions started to return, motivating them further [16, 17].

**Adjustment.** There was wide variation in participants' experiences of recovery, coping with and adjusting to their new situation. For some recovery lasted months and was full, whereas others were still experiencing residual symptoms years later [18]. Those with continuing symptoms felt they had been forced to take over the responsibility for managing and treating their own symptoms and developed personal strategies to overcome physical difficulties, such as walking more slowly to save energy or using hydrotherapy to strengthen and relax their bodies [18]. During recovery, many participants described their need for increased control and independence [20], which they associated with improved physical capabilities [16]. Achieving independence was inspiring to participants and, together with their inner strength, were the key factors in gaining full recovery [16, 17, 20].

Setting and achieving 'milestones' was another major motivating factor, with different participants viewing different points in their patient journey as milestones [16, 17, 19, 20]. For some, it was being able to walk again [17] or becoming independent [20], while for others it was moving to the rehabilitation ward [16] or going back to work [19].

Adjusting to their new situation required participants to first accept their new circumstances [18–20] and this was a complicated process for many, with some neglecting the influence of the consequences of the disease on their life situation, and others reappraising their new situation and trying to find new ways to manage their residual difficulties [18]. For some participants that meant remaining positive, often despite persisting symptoms [18, 20], or choosing to focus on the positive prognosis, and not thinking about a possible negative outcome (i.e. not recovering fully) [17].

**Towards a new self.** A recurring theme was participants' attempts and eagerness to return to their 'normal' pre-GBS selves and everyday lives. Often, going back to an acceptably

'normal' identity and avoiding the potential stigma of GBS required rejection of overt disability, with participants trying to conceal their impairments or avoid discussing them [19], especially once they were back to work [19]. As a result, participants sometimes did not inform current or prospective employers about their residual difficulties [19].

Participants were sensitive to other's reactions at work, especially when colleagues' behaviour changed towards them [19]. In order to maintain their identity and cope with threats to their self-image, different coping strategies were used. One participant went as far as entering a 'supernormal' phase on his return to work, doing as much work as possible, and refusing extra help [19]. Others monitored colleagues' behaviour for possible negative responses, and sought to exert some control over these, for example by using humour [19]. And for one participant, it was having a senior position in the company and positive relationships with colleagues that were the contributory factors that helped him be open about his diagnosis and residual difficulties [19].

Despite these concerns, returning to work was seen positively, as going back to their 'normal' selves again, and offered a distraction from the participants' residual difficulties [19]. Reconnecting with colleagues was viewed as another benefit [19]. Other motivations for going back to work were financial, recovering physically, and having a purpose and structure to everyday life [19]. Some participants, however, reported feeling vulnerable returning to work, as they were worried that discussing their health difficulties would sound like complaining [19].

The main factor facilitating participants' return to work was workplace adaptations. For some, such accommodations included having the right workplace resources or limiting their responsibilities, especially if they considered stress as a factor in their illness, by being able to change or reduce working hours [19]. For those who were more senior in their workplace, modifying their role was even easier [19].

Some workplaces did not have any appropriate resources, such as a quiet place for resting, and when participants went back to work, they were offered only short term flexibility, being expected to soon fully return to their previous responsibilities [19]. Furthermore, not all participants considered work adaptations positively, making them feel vulnerable and threatening their re-established, pre-illness, 'normal' selves [19]. For others, accepting continued support went against their personal values or diminished their sense of achievement [19].

Overall, it was evident from all five studies that living with GBS had been a life-changing experience [16–20], often making participants re-evaluate or change their life accordingly [18].

## Discussion

This systematic review and thematic meta-synthesis explored patients' experiences and perceptions of GBS at diagnosis, discharge and during recovery. Participants' experiences of recovery varied significantly, many still experiencing residual (physical, psychological and social) difficulties even years after their discharge from hospital. These results are closely aligned with concepts suggested by both the Illness Trajectory Framework (ITF), proposed by Corbin & Strauss [10], and Taylor's [11] theory of cognitive adaptation to threatening events.

Our results showed that the *trajectory* of GBS was uncertain and varied significantly from patient to patient, depending on the severity of their condition, the care and support they received, and their own attitudes towards recovery [11]. During the first phase of *pre-trajectory* and the *trajectory onset*, delayed diagnosis and/or misdiagnosis were often reported, a finding that has been well documented previously. For example, Dubey and colleagues [21] reported that on initial emergency department visit, GBS was suspected in only 49.3% of the cases; this

increased significantly when the patient was evaluated by a neurologist (67.5%) rather than the emergency department physician.

Although the *crisis and acute phases* left the participants fighting for their lives, once their condition had stabilised (*stable phase*), participants slowly started to recover. However, not all participants recovered fully and GBS continued affecting every aspect of their life, leaving them feeling lost in a changing, and often incomprehensible, life (*unstable phase*). These results echo those found in previous studies [6, 7, 9], which have also documented GBS' lasting effects.

There are many factors that affect how an illness progresses and how recovery can be affected positively or negatively. Our results, for example, showed that patients struggled with the lack of knowledge among healthcare professionals and the lack of information about GBS they received. Similar results were also found in past studies [8]. Our findings also showed that participants felt that their needs were not being met, mainly due to fractured care, lack of continuity and of personalised care. Uprichard and colleagues [22] also highlighted the importance of, and need for, an individualised approach to care for patients with GBS and how such care is essential for reducing the traumatic experience of recovery from such a severe illness.

Another facilitator to recovery was support from family and friends, also found in past studies [23]. Seeing how past GBS patients were able to get better and recover successfully (peer support) was also beneficial, by providing patients with models of good adjustment and giving them hope for their recovery and their future. One possible mechanism of having achieved this may have been by making upward social comparisons to others who were doing better than they were, thus, contributing to their self-enhancement and helping restore their self-esteem [11]. Past studies with patients in recovery from stroke have also documented the importance of support from stroke survivors in helping patients feel empowered, encouraged, motivated, validated, and less alone [24, 25].

More importantly, our meta-synthesis found novel factors that might positively affect recovery. Maintaining a positive attitude, for example, was the first step towards recovery from GBS and helped participants take control of their situation, try to manage their symptoms themselves and regain their independence; a finding in accordance with Taylor's theory [11], which argues that one of the main pathways to adjustment is by gaining a feeling of control over the threatening event.

Achieving major milestones also helped participants adjust to and *come to terms* with their new situation. Returning to work was especially seen as a really important step in going back to their 'normal' selves again, even though sometimes participants had to make accommodations before being able to return to their previous role. Such accommodations were necessary due to residual physical difficulties, but can also be seen as an attempt to control the situation by making changes in their lives that would allow them to adapt to their new situation successfully [11]. Being diagnosed with and surviving GBS was a life-changing experience for all participants that often made them search for meaning in their new situation, re-appraise their lives and re-order their priorities [11].

For the majority of GBS cases there is no real *downward* or *dying phase*; mortality in GBS patients varies widely with rates between 1–18% [26]. However, the experience of illness will be present until death.

## Strengths and limitations

This review has brought together papers discussing different aspects of people's experience (such as during an acute episode [20] or the initial phase of GBS [17], the recovery phase [16, 18], and their experiences returning to work [19]) and has synthesised them for the first time

into a comprehensive overview of the illness and recovery journey of people diagnosed with GBS.

The study followed a rigorous pre-specified protocol (registered with PROSPERO), which ensured that the review process was transparent and replicable. We conducted a comprehensive search for published and unpublished work, through twelve electronic databases, internet searches and scanning of bibliographies. The five included studies were of acceptable quality and included rich data.

A potential limitation of this review was including exclusively English-language papers, as important evidence may have been excluded due to language restrictions. However, methods for translating concepts across languages, in addition to the initial challenge of translating them across studies, have not been sufficiently developed [27]. Finally, all five studies interviewed people living with GBS, but none included other variants of GBS, such as CIDP or Miller-Fisher syndrome. Future studies should also aim to include participants diagnosed with other variants of GBS, as these sub-groups of participants may have different experiences and needs.

## Implications for policy and practice

Exploring this literature has enabled us to identify how patients may need extra support to cope better with their recovery and also identify ways that healthcare professionals and services can help facilitate further such a recovery.

One of the most important areas that needs to be addressed is the lack of knowledge about GBS among many healthcare professionals, including the lack of provision of information to patients about their condition and prognosis. Offering additional training on GBS for healthcare professionals might be an appropriate first step towards improving their knowledge, while providing educational resources and information for the public could be another helpful action. Informing patients of available support services (such as financial aid, health and social care services, as well as relevant charities) would further ensure that people receive appropriate and personalised care, facilitate their transition from hospitalisation to returning to their everyday lives and, potentially, aid recovery.

Patients' psychological needs were often not met by healthcare services, while maintaining a positive attitude was identified as essential for participants to be able to cope with and successfully recover from GBS. It would be useful, therefore, to add psychological therapies to patients' treatment regimens, if needed and wanted by patients.

We found that participants also viewed peer support as important in their road to recovery. Peer support could help address both areas discussed earlier, regarding lack of information and emotional support offered to patients with GBS. This could also be an area for future research, exploring how peer support should be provided and that it was clinically beneficial before planning for peer support to be widely available to patients with GBS, potentially through hospitals or GBS charities.

Finally, patients often reported requiring extra support to enable them going back to work. This, according to our results, would include increased awareness of GBS and its sequelae for employers, which in turn would increase understanding of the condition by employers and, therefore, making them more inclined to provide adaptations at work for patients wanting to return to their jobs (e.g. flexible working hours/days, areas for rest, etc.).

## Conclusions

This systematic meta-synthesis explored patients' experiences of GBS at diagnosis, discharge and during recovery. One factor that positively influenced management and eventually

outcomes was having a positive attitude and thinking towards recovery. Other key factors influencing management were receiving adequate information about GBS, having support from valued others (such as family members, friends or peer support), and receiving satisfactory care from healthcare services (especially nursing care). Despite the variety of experiences, it was evident from all included studies that being diagnosed with and surviving GBS was a life-changing experience for all participants.

## Supporting information

**S1 File. ENTREQ checklist.**
(DOCX)

**S2 File. PROSPERO protocol registration.**
(PDF)

## Acknowledgments

We would like to thank the members of the Community and Health Research Unit (CaHRU) study review group (University of Lincoln) for their valuable comments on a draft of this paper.

## Author Contributions

**Conceptualization:** Despina Laparidou, Ffion Curtis, Joseph Akanuwe, Jennifer Jackson, Timothy L. Hodgson, A. Niroshan Siriwardena.

**Formal analysis:** Despina Laparidou, Ffion Curtis, Joseph Akanuwe, Jennifer Jackson, Timothy L. Hodgson, A. Niroshan Siriwardena.

**Funding acquisition:** Jennifer Jackson, Timothy L. Hodgson, A. Niroshan Siriwardena.

**Investigation:** Despina Laparidou, Ffion Curtis, Joseph Akanuwe.

**Methodology:** Despina Laparidou, Ffion Curtis, Joseph Akanuwe, Jennifer Jackson, Timothy L. Hodgson, A. Niroshan Siriwardena.

**Project administration:** Despina Laparidou.

**Supervision:** A. Niroshan Siriwardena.

**Writing – original draft:** Despina Laparidou.

**Writing – review & editing:** Ffion Curtis, Joseph Akanuwe, Jennifer Jackson, Timothy L. Hodgson, A. Niroshan Siriwardena.

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
