## [Decision Letter · Decision Letter 0]

27 Apr 2020

PONE-D-19-33700

Patients’ experiences and perceptions of Guillain-Barré syndrome: a systematic review and meta-synthesis of qualitative research

PLOS ONE

Dear Dr Aloysius Niroshan Siriwardena

Thank you for submitting your manuscript to PLOS ONE. After careful consideration, we feel that it has merit but does not fully meet PLOS ONE’s publication criteria as it currently stands. Therefore, we invite you to submit a revised version of the manuscript that addresses the points raised during the review process.

We would appreciate receiving your revised manuscript by 30 days. To enhance the reproducibility of your results, we recommend that if applicable you deposit your laboratory protocols in protocols.io, where a protocol can be assigned its own identifier (DOI) such that it can be cited independently in the future. For instructions see: http://journals.plos.org/plosone/s/submission-guidelines#loc-laboratory-protocols

We look forward to receiving your revised manuscript.

Kind regards,

Federico Bilotta

Academic Editor

PLOS ONE

2. We note that your search concludes in November 2018.

It is not clear why studies published in the last 12 months were not included for consideration.

Please provide a rationale for the search window, or update the search to include studies from the last 12 months.

Additional Editor Comments:

PONE-D-19-33700

In this SR and meta-synthesis, the Authors have discussed the patients’ experiences and perceptions of Guillain-Barré syndrome (GBS) and its variants at diagnosis, discharge and during recovery, by conducting a systematic review and thematic meta-synthesis of qualitative studies of patients’ experiences of GBS (and its variants).

The Authors searched 12 electronic databases, supplemented with internet searches and forward and backward citation tracking from the included studies and review articles. Data were synthesised thematically following the Thomas and Harden approach.

The Authors identified a total of 4,204 citations and after removing duplicates and excluding citations based on title and abstract, and full-text screening, five studies were included in the review and meta-synthesis; all included studies were considered of acceptable quality. Through constant discussions and an iterative approach, we developed six analytical themes following a patients journey from suspecting that they had a health problem, through to being hospitalised, experiencing ongoing difficulties, slowly recovering from GBS, adjusting to their new circumstances, and re-evaluating their lives.

The Authors concluded that, that being diagnosed with and surviving GBS was a life-changing experience for all participants.

Editor’s comments: this SR is interesting, but provide some methodological limitations.

The main outcome of the study should be summarize as last sentence of the introduction.

The subchapter “Theoretical perspective” should be shortened by 50%, the introduction section is lengthily and not focused on the primary end point.

The Table should be reported at the end of the manuscript, and cited in the text.

The Result section is composed by 20 pages. The Section is too lengthily. Please report only the results statistically significant in the text, the all results should be reported only in the Table.

The first sentence of the Discussion section should summarize the main results of the study.

Also the Discussion section is long and lengthily. Please shortened by 30 to 40% and focused it on the primary end points of the study.

Reviewer 1: I've read with great interested this work by Siriwardena et al. dedicated to patients’ perceptions of Guillain-Barré syndrome. Undoubtedly, the data presented in this article are of great interest for specialists working with such patients. However, in my opinion, this article is extremely overloaded with data, which makes it really huge and looks like small book or thesis but not journal article. For instance, discussion section essentially repeats, in an abbreviated version, most of the theses set out in the results.

The design of the study is not completely looks optimal to me, because it is not clear what new information this meta-synthesis provides us, compared to the five qualitative studies that were included in the analysis. What has changed in our understanding of the problem after this work? It is also not entirely clear why the authors excluded quantitative works, because, at least, discussion sections of them could also contain some useful information.

In my opinion, the work needs a serious revision aimed at systematization and more concise presentation of data. Major revision

Reviewer 2: I enjoyed reading this review and found many useful features. I particularly appreciated its attempt to engage with underlying theory regarding disease trajectories.

Overall the review is sound and has the potential to make an important contribution to understanding. Unfortunately there are some elementary miscarriages in its execution that would need addressing before publication. Some of the minor issues are listed in more detail below but for ease of identification here are my major concerns:

1. The authors confuse Reporting Standards with Standards of Conduct. Following good reporting standards does not ensure a good review and so it would be good to only cite reporting standards to affirm what they are intended for.

2. Certain points betray gravitation towards a quantitative paradigm e.g. discussion of "risk of bias", "blinding to authors", numbers of participants supporting a finding and numbers of studies required in a qualitative synthesis. Particularly missing (to address some of these) is a "reflexive statement" see example below given from this same target journal. This will also address potential influence of the charity as funder (see below)..

3. The authors present a "bricolage" of findings in the Results. Not only does it engage with the conventions of quantitative research (e.g. "most" participants and "many" participants) but it attributes specific individual statements to groups of individuals misrepresenting an individually worded comment as if an agreed common experience. The whole Results section needs rewriting framing findings as overall statements followed by illustrative verbatim extracts (judiciously selected). In practical terms the authors should look at each verbatim extract and ask: does this capture a common experience? can I frame these related extracts in my own words as generalities, or is this a unique insight? does this wording express the experience better than I can myself? At the moment these results are knitted togeher from the reviewers, the source authors and the verbatim extracts of participants.

4. Although studies are not excluded on the basis of quality it would be useful to at least comment on CASP questions that the collective set of five studies performed less well against (typically reflexivity performs less well).

Specific comments:

“Due to the rarity of this condition there is limited evidence exploring the experiences of individuals who have had GBS.” This sentence perpetrates a common fallacy i.e. that the prevalence of research is linked to prevalence of a condition. In many cases the reverse is true (and this has been particularly recorded for neurological disorders) ie. That rare conditions are more attractive to research than common ones. This is partly attributable to the fact that journals are part scientific documentation and part journalism, partly to the “politics” of research funding and partly to other factors.

“CIDP” – this appears in the Review question without being written in full. The review question should have both conditions in full with respective abbreviations in brackets. (Otherwise CIDP is used twice BEFORE being written in full). It is not appropriate to use unexplained abbreviations in the review question.

“January 2000 and November 2018, to ensure relevance to the present day,” This reads as a weak justification for date limit (cynically, I would say that you have probably selected an easily memorable but topically meaningless start date!).

“Joanna Briggs Institute” – It is unclear what this Data Source is.

“The search terms were entered in all possible combinations using Boolean operators and truncation, wherever deemed necessary” Not true! They would have been kept in sets of related concepts i.e. the GBS concept, the qualitative research concept. Not ALL possible combinations. Search terms should also best be presented as two sets for greater clarity. E.g. “(i) Guillain-Barré syndrome (GBS), chronic inflammatory demyelinating polyneuropathy (CIDP), acute inflammatory demyelinating polyneuropathy (AIDP) combined with (ii) qualitative research, interview, focus group, experiences, perceptions, attitudes, and views”.

“All studies were reviewed and screened by three reviewers” This is very unclear. Does this mean that “all references were reviewed and screened”. In a review “studies” don’t become “studies” until they are included. Before that they are first “references” and then “(full text) papers”

“Qualitative Checklist [15] was used to assess risk of bias in included studies of this review” – The CASP checklist is not validated for “risk of bias”. Indeed what “risk of bias” means in a qualitative sense is epistemologically unclear anyway. Describe it only as “quality assessment”.

“Quality, however, was not a criterion for exclusion of a study” This is a sound decision but requires a sentence or clause of explanation e.g. “given that the intent was interpretative….” Or something similar. Also, being accurate “quality” is not a criterion. “Poor quality” would be.

“Blinding to the names of the authors, journals, and results of each study was not possible,

since the same authors performed the screening and data extraction of the studies to be

included in the meta-synthesis.” You don’t typically see this spelled out, particularly as you don’t explain why this could be an issue. In my view this is a very quantitative way of handling a legitimate concern. In a QUALITATIVE synthesis like this I would expect instead a reflexivity statement that talks about all the authors positionality to the data.

See the following example from PLOS One:

“Reflexive statement. Reflexive accounting allows the reader of the final research product to assess the degree to which the prior views and experiences of the researcher may have influenced the design, data collection and data interpretation of the study or in this case, the synthesis of the findings of multiple studies. This review was conceived with an informed knowledge of caesarean section and a degree of professional distance, which arguably limited bias based on the team’s own experiences. APB is a medical officer with over 15 years of experience in maternal and perinatal health research and public health in general, and caesarean section in particular. CK, a medical sociologist, came to the project with prior beliefs about the complexity and interdependency of social factors driving caesarean section rates, principally informed Stakeholder views of interventions to reduce unnecessary caesareans targeted at organisations and systems by undertaking earlier primary research with women and health professionals in the UK. SD, a Professor of Midwifery, believed that maternity care organisations are complex adaptive systems, and that the organisational ethos can exert either toxic or enhancing effects that have real consequences for staff morale, engagement, attitudes, behaviours and performance”.

https://journals.plos.org/plosone/article/file?type=printable&id=10.1371/journal.pone.0203274

“Initially, participants from most included studies” Avoid this quantitative reporting , especially when you then attribute multiple studies to a single verbatim extract as in this example. This should be reported as:

“Participants variously described initial experiences associated with their first awareness of their condition: “strange or odd sensations or peculiar feelings”, such as tingling in their feet or having difficulty opening the lid of orange juice containers etc

Similarly with “Most participants, “tried to ignore the strangeness of their bodies” whereas”. Especially be aware that “most participants” or “many participants” from one study may not be “many” or “most” across the five studies. This is particularly important because an experience is not judged important or not simply on the basis of whether it is a common experience or unique to that individual.

“Given that not many (especially not qualitative) studies were found that incorporated the

experiences of GBS from onset of symptoms and onwards” – this sentence is difficult to interpret – by using words such as Qualitative/focus groups etcetera in your search strategy you were only looking for QUALITATIVE studies. The fact that you retrieved some that weren’t qualitative isn’t a valid basis for making pronouncements about the quantitative studies which would not be representative of the ones you did not retrieve. Focus only on what you were studying, not what you accidentally retrieved along the way.

“well established guidelines (ENTREQ), which limited the potential for bias” You make the mistake here (and above in Methods) of assuming that following ENTREQ REPORTING guidelines somehow has an impact on the quality of the CONDUCT of your review. What reporting guidelines do is make it easier for a Reader to detect threats to bias (however that can be defined within a qualitative paradigm); they don’t make your review better.

“A potential limitation of this review was only five studies met the inclusion criteria for meta

synthesis” – this is largely an irrelevancy – you should consider the adequacy of your data (cp GRADE CERQual on adequacy) which is a function of both numbers of studies and their richness. So reframe as adequacy – they could still be adequate even if not plentiful. If, however they are neither rich nor plentiful then you should specifically identify what data were missing.

Similarly the reason why there are no guidelines on how many studies should be included is that this is bringing in a quantitative criterion to a qualitative phenomenon. Why are MORE studies needed? Not because the numbers aren’t sufficient – if you are going to recommend more studies then you must state which perspectives or elements of the experience are missing and therefore need more data. Otherwise you are simply advocating wasted research monies!

“Including exclusively English-language papers could be considered a weakness, as

important evidence may have been excluded due to language restrictions” This is true but you might wish to rebut this by saying “However, methods for translating concepts across languages, in addition to the initial challenge of translating them across studies, have not been sufficiently developed”.

There are several mentions of the role of “charities” in the Discussion. Given the funding for this study it would be good for the Reflexivity statement (as recommended above) to specifically address this. E.g. “The authors consider provision of information and support from charities as an important resource for patients, particularly with rare conditions, but such support may be available from multiple sources, and our view on potential sources may have been influenced by the funding for this study”. Major revision

Reviewers' comments:

Reviewer's Responses to Questions

**Comments to the Author**

1. Is the manuscript technically sound, and do the data support the conclusions?

Reviewer #1: Partly

Reviewer #2: Partly

2. Has the statistical analysis been performed appropriately and rigorously? 

Reviewer #1: Yes

Reviewer #2: N/A

3. Have the authors made all data underlying the findings in their manuscript fully available?

Reviewer #1: Yes

Reviewer #2: Yes

4. Is the manuscript presented in an intelligible fashion and written in standard English?

Reviewer #1: Yes

Reviewer #2: Yes

5. Review Comments to the Author

Reviewer #1: I've read with great interested this work by Siriwardena et al. dedicated to patients’ perceptions of Guillain-Barré syndrome. Undoubtedly, the data presented in this article are of great interest for specialists working with such patients. However, in my opinion, this article is extremely overloaded with data, which makes it really huge and looks like small book or thesis but not journal article. For instance, discussion section essentially repeats, in an abbreviated version, most of the theses set out in the results.

The design of the study is not completely looks optimal to me, because it is not clear what new information this meta-synthesis provides us, compared to the five qualitative studies that were included in the analysis. What has changed in our understanding of the problem after this work? It is also not entirely clear why the authors excluded quantitative works, because, at least, discussion sections of them could also contain some useful information.

In my opinion, the work needs a serious revision aimed at systematization and more concise presentation of data.

Reviewer #2: I enjoyed reading this review and found many useful features. I particularly appreciated its attempt to engage with underlying theory regarding disease trajectories.

Overall the review is sound and has the potential to make an important contribution to understanding. Unfortunately there are some elementary miscarriages in its execution that would need addressing before publication. Some of the minor issues are listed in more detail below but for ease of identification here are my major concerns:

1. The authors confuse Reporting Standards with Standards of Conduct. Following good reporting standards does not ensure a good review and so it would be good to only cite reporting standards to affirm what they are intended for.

2. Certain points betray gravitation towards a quantitative paradigm e.g. discussion of "risk of bias", "blinding to authors", numbers of participants supporting a finding and numbers of studies required in a qualitative synthesis. Particularly missing (to address some of these) is a "reflexive statement" see example below given from this same target journal. This will also address potential influence of the charity as funder (see below)..

3. The authors present a "bricolage" of findings in the Results. Not only does it engage with the conventions of quantitative research (e.g. "most" participants and "many" participants) but it attributes specific individual statements to groups of individuals misrepresenting an individually worded comment as if an agreed common experience. The whole Results section needs rewriting framing findings as overall statements followed by illustrative verbatim extracts (judiciously selected). In practical terms the authors should look at each verbatim extract and ask: does this capture a common experience? can I frame these related extracts in my own words as generalities, or is this a unique insight? does this wording express the experience better than I can myself? At the moment these results are knitted togeher from the reviewers, the source authors and the verbatim extracts of participants.

4. Although studies are not excluded on the basis of quality it would be useful to at least comment on CASP questions that the collective set of five studies performed less well against (typically reflexivity performs less well).

Specific comments:

“Due to the rarity of this condition there is limited evidence exploring the experiences of individuals who have had GBS.” This sentence perpetrates a common fallacy i.e. that the prevalence of research is linked to prevalence of a condition. In many cases the reverse is true (and this has been particularly recorded for neurological disorders) ie. That rare conditions are more attractive to research than common ones. This is partly attributable to the fact that journals are part scientific documentation and part journalism, partly to the “politics” of research funding and partly to other factors.

“CIDP” – this appears in the Review question without being written in full. The review question should have both conditions in full with respective abbreviations in brackets. (Otherwise CIDP is used twice BEFORE being written in full). It is not appropriate to use unexplained abbreviations in the review question.

“January 2000 and November 2018, to ensure relevance to the present day,” This reads as a weak justification for date limit (cynically, I would say that you have probably selected an easily memorable but topically meaningless start date!).

“Joanna Briggs Institute” – It is unclear what this Data Source is.

“The search terms were entered in all possible combinations using Boolean operators and truncation, wherever deemed necessary” Not true! They would have been kept in sets of related concepts i.e. the GBS concept, the qualitative research concept. Not ALL possible combinations. Search terms should also best be presented as two sets for greater clarity. E.g. “(i) Guillain-Barré syndrome (GBS), chronic inflammatory demyelinating polyneuropathy (CIDP), acute inflammatory demyelinating polyneuropathy (AIDP) combined with (ii) qualitative research, interview, focus group, experiences, perceptions, attitudes, and views”.

“All studies were reviewed and screened by three reviewers” This is very unclear. Does this mean that “all references were reviewed and screened”. In a review “studies” don’t become “studies” until they are included. Before that they are first “references” and then “(full text) papers”

“Qualitative Checklist [15] was used to assess risk of bias in included studies of this review” – The CASP checklist is not validated for “risk of bias”. Indeed what “risk of bias” means in a qualitative sense is epistemologically unclear anyway. Describe it only as “quality assessment”.

“Quality, however, was not a criterion for exclusion of a study” This is a sound decision but requires a sentence or clause of explanation e.g. “given that the intent was interpretative….” Or something similar. Also, being accurate “quality” is not a criterion. “Poor quality” would be.

“Blinding to the names of the authors, journals, and results of each study was not possible,

since the same authors performed the screening and data extraction of the studies to be

included in the meta-synthesis.” You don’t typically see this spelled out, particularly as you don’t explain why this could be an issue. In my view this is a very quantitative way of handling a legitimate concern. In a QUALITATIVE synthesis like this I would expect instead a reflexivity statement that talks about all the authors positionality to the data.

See the following example from PLOS One:

“Reflexive statement. Reflexive accounting allows the reader of the final research product to assess the degree to which the prior views and experiences of the researcher may have influenced the design, data collection and data interpretation of the study or in this case, the synthesis of the findings of multiple studies. This review was conceived with an informed knowledge of caesarean section and a degree of professional distance, which arguably limited bias based on the team’s own experiences. APB is a medical officer with over 15 years of experience in maternal and perinatal health research and public health in general, and caesarean section in particular. CK, a medical sociologist, came to the project with prior beliefs about the complexity and interdependency of social factors driving caesarean section rates, principally informed Stakeholder views of interventions to reduce unnecessary caesareans targeted at organisations and systems by undertaking earlier primary research with women and health professionals in the UK. SD, a Professor of Midwifery, believed that maternity care organisations are complex adaptive systems, and that the organisational ethos can exert either toxic or enhancing effects that have real consequences for staff morale, engagement, attitudes, behaviours and performance”.

https://journals.plos.org/plosone/article/file?type=printable&id=10.1371/journal.pone.0203274

“Initially, participants from most included studies” Avoid this quantitative reporting , especially when you then attribute multiple studies to a single verbatim extract as in this example. This should be reported as:

“Participants variously described initial experiences associated with their first awareness of their condition: “strange or odd sensations or peculiar feelings”, such as tingling in their feet or having difficulty opening the lid of orange juice containers etc

Similarly with “Most participants, “tried to ignore the strangeness of their bodies” whereas”. Especially be aware that “most participants” or “many participants” from one study may not be “many” or “most” across the five studies. This is particularly important because an experience is not judged important or not simply on the basis of whether it is a common experience or unique to that individual.

“Given that not many (especially not qualitative) studies were found that incorporated the

experiences of GBS from onset of symptoms and onwards” – this sentence is difficult to interpret – by using words such as Qualitative/focus groups etcetera in your search strategy you were only looking for QUALITATIVE studies. The fact that you retrieved some that weren’t qualitative isn’t a valid basis for making pronouncements about the quantitative studies which would not be representative of the ones you did not retrieve. Focus only on what you were studying, not what you accidentally retrieved along the way.

“well established guidelines (ENTREQ), which limited the potential for bias” You make the mistake here (and above in Methods) of assuming that following ENTREQ REPORTING guidelines somehow has an impact on the quality of the CONDUCT of your review. What reporting guidelines do is make it easier for a Reader to detect threats to bias (however that can be defined within a qualitative paradigm); they don’t make your review better.

“A potential limitation of this review was only five studies met the inclusion criteria for meta

synthesis” – this is largely an irrelevancy – you should consider the adequacy of your data (cp GRADE CERQual on adequacy) which is a function of both numbers of studies and their richness. So reframe as adequacy – they could still be adequate even if not plentiful. If, however they are neither rich nor plentiful then you should specifically identify what data were missing.

Similarly the reason why there are no guidelines on how many studies should be included is that this is bringing in a quantitative criterion to a qualitative phenomenon. Why are MORE studies needed? Not because the numbers aren’t sufficient – if you are going to recommend more studies then you must state which perspectives or elements of the experience are missing and therefore need more data. Otherwise you are simply advocating wasted research monies!

“Including exclusively English-language papers could be considered a weakness, as

important evidence may have been excluded due to language restrictions” This is true but you might wish to rebut this by saying “However, methods for translating concepts across languages, in addition to the initial challenge of translating them across studies, have not been sufficiently developed”.

There are several mentions of the role of “charities” in the Discussion. Given the funding for this study it would be good for the Reflexivity statement (as recommended above) to specifically address this. E.g. “The authors consider provision of information and support from charities as an important resource for patients, particularly with rare conditions, but such support may be available from multiple sources, and our view on potential sources may have been influenced by the funding for this study”.

6. PLOS authors have the option to publish the peer review history of their article (what does this mean?). If published, this will include your full peer review and any attached files.

Reviewer #1: No

Reviewer #2: Yes: Andrew Booth

---

## [Author Response · Author response to Decision Letter 0]

22 May 2020

See uploaded file: Response to reviewers

---

## [Decision Letter · Decision Letter 1]

17 Nov 2020

PONE-D-19-33700R1

Patients’ experiences and perceptions of Guillain-Barré syndrome: a systematic review and meta-synthesis of qualitative research

PLOS ONE

Dear Dr. Siriwardena,

Thank you for submitting your manuscript to PLOS ONE. After careful consideration, we invite you to undertake some minor revision and submit a revised version of the manuscript that addresses the points raised during the review process.

We look forward to receiving your revised manuscript.

Kind regards,

Kathleen Finlayson

Academic Editor

PLOS ONE

Additional Editor Comments (if provided):

Thank you for addressing the reviewers' comments, the review is well written and addresses a significant area of research. Please note the recommendation below to further condense the article. I would recommend this for the introduction, and adding in the recommendations for policy and practice within the discussion.

Reviewers' comments:

Reviewer's Responses to Questions

**Comments to the Author**

1. If the authors have adequately addressed your comments raised in a previous round of review and you feel that this manuscript is now acceptable for publication, you may indicate that here to bypass the “Comments to the Author” section, enter your conflict of interest statement in the “Confidential to Editor” section, and submit your "Accept" recommendation.

Reviewer #1: (No Response)

2. Is the manuscript technically sound, and do the data support the conclusions?

Reviewer #1: Partly

3. Has the statistical analysis been performed appropriately and rigorously? 

Reviewer #1: N/A

4. Have the authors made all data underlying the findings in their manuscript fully available?

Reviewer #1: Yes

5. Is the manuscript presented in an intelligible fashion and written in standard English?

Reviewer #1: Yes

6. Review Comments to the Author

Reviewer #1: I thank authors for their efforts to improve the paper but from my point of view manuscript is still too long for publication as a journal article, especially it important for intro and results sections.

7. PLOS authors have the option to publish the peer review history of their article (what does this mean?). If published, this will include your full peer review and any attached files.

Reviewer #1: No

---

## [Author Response · Author response to Decision Letter 1]

18 Dec 2020

Dear Dr Bilotta, 

Article title: Patients’ experiences and perceptions of Guillain-Barré syndrome: a systematic review and meta-synthesis of qualitative research

Thank you for your further feedback on the paper and advice from editors and reviewers on minor revision of the paper following our previous major revision. 

We have substantially shorted the introduction and results and adjusted the discussion in line with the advice given. Details of our response are below.

Thank you for reconsidering the revised paper for publication. 

Yours sincerely,

Prof A. N. Siriwardena on behalf of the authors

 

Editor/reviewer comments Response

Additional Editor Comments (if provided): Thank you for addressing the reviewers' comments the review is well written and addresses a significant area of research. Please note the recommendation below to further condense the article. I would recommend this for the introduction and adding in the recommendations for policy and practice within the discussion. Thank you. We have condensed the text considerably and added recommendations for policy and practice in the discussion.

Reviewer #1: I thank authors for their efforts to improve the paper but from my point of view manuscript is still too long for publication as a journal article especially it important for intro and results sections. 

Thank you. We have shortened the article as far as possible without changing the meaning of the text. The Background has been reduced from 1042 to 893 words and the Results from 2528 to 2103 words. The other sections

---

## [Editor Report · Decision Letter 2]

11 Jan 2021

Patients’ experiences and perceptions of Guillain-Barré syndrome: a systematic review and meta-synthesis of qualitative research

PONE-D-19-33700R2

Dear Dr. Siriwardena,

We’re pleased to inform you that your manuscript has been judged scientifically suitable for publication and will be formally accepted for publication once it meets all outstanding technical requirements.

Kind regards,

Kathleen Finlayson

Academic Editor

PLOS ONE

Additional Editor Comments (optional):

Thank you for your response to the reviewers' suggestions. The manuscript is more succinct and provides a valuable contribution to understanding of this area.
---

## [Editor Report · Acceptance letter]

13 Jan 2021

PONE-D-19-33700R2 

Patients’ experiences and perceptions of Guillain-Barré syndrome: a systematic review and meta-synthesis of qualitative research 

Dear Dr. Siriwardena:

I'm pleased to inform you that your manuscript has been deemed suitable for publication in PLOS ONE. Congratulations! Your manuscript is now with our production department. 

Kind regards, 

on behalf of

Dr. Kathleen Finlayson 

Academic Editor

PLOS ONE